# Surprising Negative Results for Generative Adversarial Tree Search

## Abstract

While many recent advances in deep reinforcement learning rely on model-free methods, model-based approaches remain an alluring prospect for their potential to exploit unsupervised data to learn environment dynamics. One prospect is to pursue hybrid approaches, as in AlphaGo, which combines Monte-Carlo Tree Search (MCTS)—a model-based method—with deep-$Q$ networks (DQNs)—a model-free method. MCTS requires generating rollouts, which is computationally expensive. In this paper, we propose to simulate roll-outs, exploiting the latest breakthroughs in image-to-image transduction, namely Pix2Pix GANs, to predict the dynamics of the environment. Our proposed algorithm, *generative adversarial tree search* (GATS), simulates rollouts up to a specified depth using both a GAN-based dynamics model and a reward predictor. GATS employs MCTS for planning over the simulated samples and uses DQN to estimate the $Q$-function at the leaf states. Our theoretical analysis establishes some favorable properties of GATS vis-a-vis the bias-variance trade-off and empirical results show that on 5 popular Atari games, the dynamics and reward predictors converge quickly to accurate solutions. However, GATS fails to outperform DQNs in 4 out of 5 games. Notably, in these experiments, MCTS has only short rollouts (up to tree depth 4), while previous successes of MCTS have involved tree depth in the hundreds. We present a hypothesis for why tree search with short rollouts can fail even given perfect modeling.

## 1 Introduction

The earliest and best-publicized applications of *deep reinforcement learning (DRL)* involve Atari games (Mnih et al., 2015) and the board game of Go (Silver et al., 2016), where experience is inexpensive because the environments are simulated. In such scenarios, DRL can be combined with *Monte-Carlo tree search (MCTS)* methods (Kearns et al., 2002; Kocsis & Szepesvári, 2006) for planning, where the agent executes roll-outs on the simulated environment (as far as computationally feasible) to finds suitable policies. However, for RL problems with long episodes, e.g. Go, MCTS can be very computationally expensive. In order to speed up MCTS for Go and learn an effective policy, Alpha Go (Silver et al., 2016) employs a depth-limited MCTS with the depth in the hundreds on their Go emulator and use an estimated $Q$-function to query the value of leaf nodes. However, in real-world applications, such as robotics (Levine et al., 2016) and dialogue systems (Lipton et al., 2016), collecting samples often takes considerable time and effort. In such scenarios, the agent typically cannot access either the environment model or a corresponding simulator.

Recently, *generative adversarial networks (GANs)* (Goodfellow et al., 2014) have emerged as a popular tool for synthesizing realistic-seeming data, especially for high-dimensional domains, including images and audio. Unlike previous approaches to image generation, which typically produced blurry images due to optimizing an L1 or L2 objective, GANs produces crisp images. Since theire original conception as an unsupervised method, GANs have been extended for conditional generation, e.g., generating an image conditioned on a label (Mirza & Osindero, 2014; Odena et al., 2016) or the next frame in a video given a context window (Mathieu et al., 2015). Recently, the PIX2PIX approach has demonstrated impressive results on a range of image-to-image transduction tasks(Isola et al., 2017).

In this work, we propose and analyze *generative adversarial tree search (GATS)*, a new DRL algorithm that utilizes samples from the environment to learn both a $Q$-function approximator, a near-term reward predictor, and a GAN-based model of the environment's dynamics (state transitions). Together, the dynamics model and reward predictor constitute a learned simulator on which MCTS can be performed. GATS leverages PIX2PIX GANs to learn a *generative dynamics model (*GDM*)* that efficiently learns the dynamics of the environment, producing images that agree closely with the actual observed transactions and are also visually crisp. We thoroughly study various image transduction models, arriving ultimately at a GDM that converges quickly (compared to the DQN), and appears from our evaluation to be reasonably robust to subtle distribution shifts, including some that destroy a DQN policy. We also train a reward predictor that converges quickly, achieving negligible error (over 99% accuracy). GATS bridges model-based and model-free reinforcement learning, using the learned dynamics and reward predictors to simulate roll-outs in combination with a DQN. Specifically, GATS deploys the MCTS method for planning over a bounded tree depth and uses the DQN algorithm to estimate the $Q$-function as a value for the leaf states (Mnih et al., 2015; Van Hasselt et al., 2016).

One notable aspect of the GATS algorithm is its flexibility, owing to consisting of a few modular building blocks: ($i$) *value learning*: we deployed DQN and DDQN ($ii$) *planning*: we use pure Monte Carlo sampling; ($iii$) *a reward predictor:* we used a simple 3-class classifier; ($iv$) *dynamics model*: we propose the GDM architecture. Practically, one can swap in other methods for any among these blocks and we highlight some alternatives in the related work. Thus, GATS constitutes a general framework for studying the trade-offs between model-based and model-free reinforcement learning.

## 1.1 TECHNICAL CONTRIBUTIONS

**Theoretical analysis**   We analyze the components of error in the estimation of the expected return used by GATS, and further study the trade-offs in its bias and variance. Since GATS utilizes the learned $Q$ function of DQN/DDQN in the leaf nodes of the MCTS tree, the existing errors in the $Q$-estimation decays exponentially as the depth of MCTS grows. We further study the bias in the $Q$ estimate of DQN and DDQN, where we found (empirically) that GATS with even one step look-ahead or rollout (depth one), can help to reduce the negative effects of these biases. This leads to a reduction in the sample complexity of DQN and DDQN by a factor of 2 on the game Pong. Furthermore, we develop a heuristic optimism-based strategy for GATS using the GDM. The low computation cost of Pong allows us to do an extensive study of the bias-variance of $Q$ for different model-based planning and exploration strategies.

**Experimental results**   For this work, we also developed a new OpenAI gym (Brockman et al., 2016)-like interface for the latest *Atari Learning Environment (ALE)* (Machado et al., 2017), which supports different modes and difficulties for Atari games. We study the sample complexity required by GDM and RP to adapt and transfer from one domain of the game (a mode and difficulty) to another domain (another mode and difficulty). We show that GDM and RP adapt quickly to the new mode in a few numbers of samples, while the estimated $Q$-function requires significantly more samples to adapt. We documented and open-sourced this wrapper on the latest ALE as well as the code for GDM, RP, and the GATS algorithm.

**Surprising negative results**   Despite learning environment models (GDM and RP) that converge efficiently and achieve accuracy exceeding our expectations, we are unable to improve the return of the learned policy using short rollouts (at most depth 5) on any other game besides Pong. The negative result persisted across an extensive and costly study with many different hyper-parameter settings and learning strategies, including several different strategies to use the generated frames to train the $Q$-model inspired by Sutton (1990). We put forth a hypothesis for why GATS, despite the good performance of its constituent models and its theoretical advantages, might fail in short rollout depths: in short, under this training regime, the problem may be that the $Q$-learner does not observe the outcomes of its mistakes. We also explain why in order to test our hypothesis, one might need to experiment with significantly longer rollouts, which may be prohibitively expensive (computationally) in this domain. Find a more detailed explanation of this hypothesis in Section 7.

Consider the fact that all known successes of MCTS have involved tree depth in the hundreds. For example, (Guo et al., 2014) shows for Atari games, when a plain MCTS is deployed, a depth of 300 with 1000 trajectories is required to learn a reasonable policy. This depth of MCTS on GDM

requires massive amounts of computation beyond the scale of academic research and the scope of this paper. Considering the broader enthusiasm for both model-based RL and generative adversarial networks, we believe that this study, despite its failure to advance the leaderboard, illuminates several important considerations for future work to develop tree-search and rollout based methods to combine model-based and model-free reinforcement learning.

## 2 PRELIMINARIES

An infinite horizon $\gamma$-discounted MDP $M$ is a tuple $\langle \mathcal{X}, \mathcal{A}, T, R, P_0, \gamma \rangle$, with state space $\mathcal{X}$, action space $\mathcal{A}$, and $P_0$, the distribution over the initial states. The transition kernel $T : x, a \rightarrow \Delta_x$, drives the model dynamics accompanied with $[0, 1]$-bounded reward of $R : x, a \rightarrow \Delta_r$, where $0 \leq \gamma < 1$. The agent's objective is to find a policy $\pi := \mathcal{X} \rightarrow \mathcal{A}$ that maximizes the overall expected discounted reward $\eta^* := \eta(\pi^*) = \max_\pi \lim_{N \rightarrow \infty} \mathbb{E}_\pi \left[ \sum_{t=0}^N \gamma^t r_t | x_0 \sim P_0 \right]$. By $Q_\pi(x, a) := \lim_{N \rightarrow \infty} \mathbb{E}_\pi \left[ \sum_{t=0}^N \gamma^t r_t | x_0 = x, a_0 = a \right]$, we denote the expected cumulative discounted reward under policy $\pi$ starting from state-action $x, a$. In value based RL, we aim to learn the $Q$-function in order to derive the optimal policy. In order to learn the $Q$ function, we might aim to minimize square loss for any given pair of state and action $(x, a)$,

$$\left( Q(x, a) - \mathbb{E}_\pi \left[ r + \gamma Q(x', a') | x, a \right] \right)^2 \tag{1}$$

In order to minimize the expression in Eq. 1, a double sampling is required to estimate the inner expectation. To avoid the cost of the double sampling, a common approach is to instead minimize the Bellman residual (Lagoudakis & Parr, 2003; Antos et al., 2008):

$$\mathbb{E}_\pi \left[ \left( Q(x, a) - (r + \gamma Q(x', a')) \right)^2 \Big| x, a \right]$$
$$= \left( Q(x, a) - \mathbb{E}_\pi \left[ r + \gamma Q(x', a') \Big| x, a \right] \right)^2 + \text{Var}_\pi \left( r + \gamma Q(x', a') \Big| x, a \right)$$

The Bellman residual is the sum of the expression in Eq. 1 and an additional variance term. DQN partially addresses this bias[1], by computing the target value with a separate function approximator, typically updated less frequently than the policy,

$$\mathcal{L}(Q, Q^{\text{target}}) = \mathbb{E}_\pi \left[ \left( Q(x, a) - r - \gamma Q^{target}(x', a') \right)^2 \right] \tag{2}$$

Generally, in addition to this bias, there are additional statistical biases due to limited capacity of network, optimization algorithm, model mismatch, as well as bias induced by the max operator or choice of $a'$. In the next section, we theoretically and empirically study this bias and show how GATS can address this undesirable effect. For the generative dynamic model, we propose a generic GDM which consists of a generator $G$ and a discriminator $D$, trained adversarially w.r.t. the extended conditional Wasserstein metric between two probability measures $\mathbb{P}_\varpi, \mathbb{P}_G$ conditioned on a third probability measure $\mathbb{P}$;

$$W(\mathbb{P}_\varpi, \mathbb{P}_G | \mathbb{P}) := \sup_{D \in \| \cdot \|_L} \mathbb{E}_{\varpi \sim \mathbb{P}_\varpi | \varrho, \varrho \sim \mathbb{P}}[D(\varpi | \varrho)] - \mathbb{E}_{\varpi : G(\varrho \sim \mathbb{P}, z \sim \mathcal{N}(0, I))}[D(\varpi | \varrho)] \tag{3}$$

Here, $z$ is a mean-zero unit-variance Gaussian vector random variable and $\| \cdot \|_L$ indicates the space of all 1-Lipschitz functions. In GDM, $D$ solves the interior $\sup$, while $G$'s objective is to minimize this distance and learn the $\mathbb{P}_\varpi | \varrho$ for all $\varrho$. We deploy our proposed GDM in GATS where $\mathbb{P}$ is the distribution over pairs of $\varrho : (x, a)$ in the replay buffer, and $\mathbb{P}_\varpi | \varrho$ is the distribution over the successor states $\varpi : x'$, which is the transition kernel $T(x'|x, a)$.

## 3 BIAS-VARIANCE TRADE-OFF

In the previous section, we discussed the DQN objective function, Eq. 2, which is an inherently biased estimator. In the next section, we demonstrate how big these biases can be in practice. Let $\widehat{\cdot}$ denote an estimate for a given quantity and $e_Q$ be the upper bound on estimation error in $Q$ function

---

[1] This bias vanishes in deterministic domains

(bias+variance); $|Q(x,a) - \widehat{Q}(x,a)| \le e_Q$ , $\forall x, a$. For any given rollout policy $\pi_r$, using GDM, RP, and estimated Q, the expected return is given by the following expression:

$$\xi_p(\pi_r, x) := \mathbb{E}_{\pi_r, \text{GDM,RP}}\left[\left(\sum_{h=0}^{H-1} \gamma^h \widehat{r}_h\right) + \gamma^H \max_a \widehat{Q}(x_H, a)\Big| x\right]. \tag{4}$$

Since this expectation is estimated with given GDM,RP and the estimated $Q$-function, GATS efficiently estimates this expected return without any interaction with the real environment. Let $\xi(\pi_r, x)$ denote the same quantity under the ground truth model

$$\xi(\pi_r, x) := \mathbb{E}_{\pi_r}\left[\left(\sum_{h=0}^{H-1} \gamma^h r_h\right) + \gamma^H \max_a Q(x_H, a)\Big| x\right]$$

Moreover, for the RP and GDM, where $\widehat{T}$ and $\widehat{r}$ are the estimated transition kernel and reward function [2] we assume $\forall x, x', a \in \mathcal{X}, \mathcal{A}$

$$\sum_a \left|\left(r(x,a) - \widehat{r}(x,a)\right)\right| \le e_R, \quad and, \quad \sum_{x'} \left|\left(T(x'|x,a) - \widehat{T}(x'|x,a)\right)\right| \le e_T$$

**Proposition 1** *[Model-based Model-free trade-off] If* GATS *is run to estimate the Q function using* DQN *procedure with the estimated model of the environment,* GDM *and* RP*, the deviation in estimating* $\xi_p(\pi_r, x) \ \forall \ x$ *and* $\pi_r$ *is bounded as;*

$$|\xi_p(\pi_r, x) - \xi(\pi_r, x)| \le \frac{1 - \gamma^H + H\gamma^H(1-\gamma)}{(1-\gamma)^2}e_T + \frac{1-\gamma^H}{1-\gamma}e_R + \gamma^H e_Q \tag{5}$$

*Proof in the Appendix A*

Proposition. 1 provides an insight to the contribution of each term in the error into the GATS predicted expected return $\xi_p(\pi_r, x)$. The exponential vanishing error in $Q$ estimation comes at the cost of variances in the model estimation. Therefore, the agent can choose $H$, the depth of roll-out, in such a way to minimize the estimation error through approximating the upper bound on error terms.

## 4 GENERATIVE ADVERSARIAL TREE SEARCH

We provide a more detailed description of generative adversarial tree search (Alg. 1). Generative Adversarial Tree Search (GATS) Alg. 1 is built upon DQN/DDQN and by re-using the experiences in the replay buffer it learns a reward model RP, model dynamics GDM, and $Q$-function. For planning, GATS deploys bounded-depth MCTS on the learned model (GDM and RP), instead of the real environment. It then uses the learned $Q$-function to estimate the maximum expected return at the leaf nodes Fig. 8. In order to learn the model dynamics, we propose a novel deep neural network, GDM, parameterized by $\theta^{\text{GDM}}$, demonstrating that the constituent models are sample-efficient and achieve strong predictive performance. The input to the GDM is the state (four consecutive frames) and a sequence of actions, from which GDM generates the successor frames. We train GDM by sampling mini-batches of experiences from the replay buffer. Simultaneously, we train RP, parameterized with $\theta^{\text{RP}}$, the same way.

In our basic experiments, our exploration strategy for GATS, as with DQN, consists of the $\epsilon$-greedy approach. We also propose a new optimism-based method of exploration for GATS. Throughout our experimental study, we observed that the approximated Wasserstein distance, the output of the discriminator, decreases for frequently-visited state-action experiences and stays high for rare experiences. Intuitively, for unfamiliar experiences, the generator is unable to fool the discriminator, so the Wasserstein distance between real and generated frames is high. We compute the exponent of this distance and use its inverse to construct a pseudo-count (Ostrovski et al., 2017), an adaptation of the idea of a state-action visitation count for continuous states and actions $\tilde{N}(x,a)$. In the optimism in the face of uncertainty principle (Jaksch et al., 2010), an optimistic Q-function $\tilde{Q}$ is learned for

---

[2]In the latter sections, we study $e_R$ and $e_T$, Fig. 1, observing that they are surprisingly small in practice, e.g. RP, on average, makes less than two mistakes per episode in Pong

---

**Algorithm 1** GATS (H)

---

1: Initialize parameter sets $\theta, \theta^{target}, \theta^{\text{GDM}}, \theta^{\text{RP}}$
2: Initialize replay buffer and set counter $= 0$
3: **for** episode = 1 to inf **do**
4:     **for** $t =$ to the end of episode **do**
5:         $a_t, \{(x_i, a_i, r_i, x_{i+1})\}_0^m = MCTS(x_t, H, \theta, \theta^{\text{GDM}}, \theta^{\text{RP}})$
6:         Store transition $(x_t, a_t, r_t, x_{t+1})$ and $\{(x_i, a_i, r_i, x_{i+1})\}_0^m$ in replay buffer
7:         Sample a random minibatch of transitions $(x_\tau, a_\tau, r_\tau, x_{\tau+1})$ from replay buffer
8:         $y_\tau \leftarrow \begin{cases} r_\tau & \text{for terminal } x_{\tau+1} \\ r_\tau + \max_{a'} Q(x_{\tau+1}, a'; \theta^{target}) & \text{for non-terminal } x_{\tau+1} \end{cases}$
9:         $\theta \leftarrow \theta - \eta \cdot \nabla_\theta (y_\tau - Q(x_\tau, a_\tau; \theta))^2$
10:        Update GDM, and RP
11:     **end for**
12: **end for**

---

exploration. We deploy this notion of optimism and use the pseudo-count to compute the optimistic $\tilde{Q}$

$$\tilde{Q}_\pi(x, a) = \widehat{r}(x, a) + c\sqrt{1/\tilde{N}(x, a)} + \gamma \sum_{x'} \widehat{T}(x'|x, a)\tilde{Q}_\pi(x', \pi(x')), \qquad (6)$$

where $c$ is the confidence scale constant. We can decouple Eq. 6 into the $Q$-function and confidence part, i.e. $\tilde{Q}_\pi(x, a) = Q_\pi(x, a) + C_\pi(x, a)$ where

$$C_\pi(x, a) := c\sqrt{1/\tilde{N}(x, a)} + \gamma \sum_{x'} \widehat{T}(x'|x, a)C_\pi(x', \pi(x')) \qquad (7)$$

Therefore, we learn $C$ the same way as we learn $Q$ by using DDQN. We add the learned $C$ to $\widehat{\xi}(\pi_r, x)$ for our GATS planning, i.e. $\max_\pi \{\widehat{\xi}(\pi, x) + C(\pi, x)\}$. This encourages the agent to explore the parts of state space where the GDM is not yet accurate. Since those parts of the state space correspond to less frequently visited parts of the state space, this approach can help with better exploration compared to $\varepsilon$-greedy approach.

## 5 EXPERIMENTS

We study the performance of GATS on 5 Atari games, namely *Pong, Asterix, Breakout, Crazy Climber* and *Freeway*, using the OpenAI Gym (Brockman et al., 2016). We adopt the common DQN architecture and game settings popularized by Mnih et al. (2015). For the GDM architecture (Fig. 10), we build upon the U-Net model image-to-image generator originally used in PIX2PIX (Isola et al., 2017). The GDM receives a state, sequence of actions, a Gaussian noise vector[3] and generates a predicted next state.[4] The RP is a simple model with 3 outputs, it receives the current state, action, and the successor state as input then outputs a class, one for each possible clipped reward $\{-1, 0, 1\}$. We train GDM and RP using prioritized weighted mini-batches of size 128 (more weight on recent samples), and update the two networks every 16 decision steps of GATS (4 times less frequently than the $Q$ update). We deploy GATS as a bounded-depth MCTS on the learned model[5] and use the learned $Q$ values at the leaves.

Our experiments show that with significantly fewer samples, compared to DQN training, the GDM learns the environment's dynamics and generalizes well to a test set. We also show it adapts quickly even if we change the policy or the difficulty or the mode of the domain. In order to develop the current GDM, we experimented many different model architectures for the generator-discriminator, as well as different loss functions. We compare performance visually on test samples, since the L1 and L2 losses are not good metrics for learning game dynamics as demonstrated in detail in Apx. F

---

[3] We set the noise to zero for deterministic environments.
[4] See Apx for detailed explanation on the architecture and optimization procedure.
[5] For short loggerhead in deterministic environment, we expand the whole tree.

and Fig. 11. We experiment with the PatchGAN discriminator (patch sizes 1, 16, and 70) and $L1$ loss used in PIX2PIX (Isola et al., 2017), finding that this architecture takes approximately $10\times$ more training iterations to learn game dynamics for Pong than the current GDM. This is likely since the learning game dynamics such as ball position requires the entire frame for the discriminator, more than the patch-based texture loss given by PatchGAN. We also experiment with the ACVP (Oh et al., 2015) architecture for the generator trained on L2 loss using the same hyper-parameters as specified in the original paper, and we find it also takes an order of magnitude more training iterations to learn game dynamics. We hypothesize this is because ACVP is a much larger network optimized for long term predictions, and does not take advantage of skip-connections and the discriminator loss as in GDM.

For the choice of the GAN loss, we first tried the original GAN-loss (Goodfellow et al., 2014), which is based on Jensen–Shannon distance. With this criterion, not only it is difficult to find the right parameters but also not stable enough for non-stationary domains. We did the experiments using this loss and trained for Pong while the resulting model was not stable enough for RL tasks. The training loss is sometimes unstable even for a given fixed data set. Since, Wasserstein metric provides Wasserstein distance criterion and is a more general loss in GANs, we deployed W-GAN (Arjovsky et al., 2017) for our GDM. Because W-GAN requires the discriminator to be a bounded Lipschitz function, the authors adopt gradient clipping. Using W-GAN provided improvement but still not sufficient for RL where fast and stable convergence is required. In order to improve the learning stability, parameter robustness, and quality of frames, we also tried the follow-up work on improved-W-GAN (Gulrajani et al., 2017), which adds a gradient penalty into the loss of discriminator in order to satisfy the bounded Lipschitzness. Even though it made the GDM more stable than before, it was still not sufficient due to the huge instability in the loss curve. Finally, we tried spectral normalization (Miyato et al., 2018), a recent technique that not only provides high-quality frames, but also converges quickly while the loss function stays smooth. Because of its stability, robustness to hyperparameter choices, and fast learning due spectral normalization combined with W-GAN, GDM is able to handle the change in the state distribution in RL and still preserve the frame quality. More detailed study is left to Appendix.

Fig. 1 shows the efficiency of GDM and how accurate it can generate next 9 frames just conditioning on the previous 4 frames and the trajectory of actions. We train the GDM on a replay buffer of 100,000 frames using a 3-step loss, and evaluate its performance on 8-step roll-outs on unseen 10,000 frames. We also tried the learned Q function on both generated and real consecutive frames and observed that the relative deviation is significantly small ($\mathcal{O}(10^{-2})$). Therefore as DRL methods are data hungry, we can re-use the data generated by GDM to train the $Q$-function even more. In later we study the ways we can incorporate the generated samples by GDM and RP in order to train the $Q$-function, as it is done in Dyna-Q. It worth noting that, since the GDM and RP modes adapt quickly, it is critical to come up with a strategy for how to sample from the replay buffer. Our analysis suggests sampling fresher experience with higher probability, compared to old, potentially stale samples (Fig. 6).

**Shift in Domain**     We extend our study to the case where we change the model dynamics by changing the game mode. In this case, by going from default mode to alternate mode in pong, the opponent paddle gets halved in size. We expected that in this case, where the game became easier, the DDQN agent would preserve its performance but surprisingly it gave us the most negative score possible, i.e -21 and broke. Therefore, we start fine-tuning DDQN and took 3M time steps (12M frames) to master the game again. It is worth noting that it takes DDQN 5M time steps (20M frames) to master from scratch. While DDQN appears unacceptably brittle in this scenario, GDM and RP adapt to the new model dynamics in 3k samples, which is significantly smaller (see details in F). For this study, we wrote an Gym-style wrapper for a new version of ALE which supports different modes of difficulty levels.

## 6    RELATED WORK

The exploration-exploitation trade-off is extensively studied in RL literature (Kearns & Singh, 2002; Brafman & Tennenholtz, 2003; Asmuth et al., 2009). The regret analysis of MDPs  (Jaksch et al., 2010; Bartlett & Tewari, 2009) is investigated, where the Optimism in the Face of Uncertainty (OFU) principle is applied to guarantee a high probability regret upper bound. For Partially Observable MDPs, OFU is known to provide a high probability regret upper bound (Azizzadenesheli et al.,

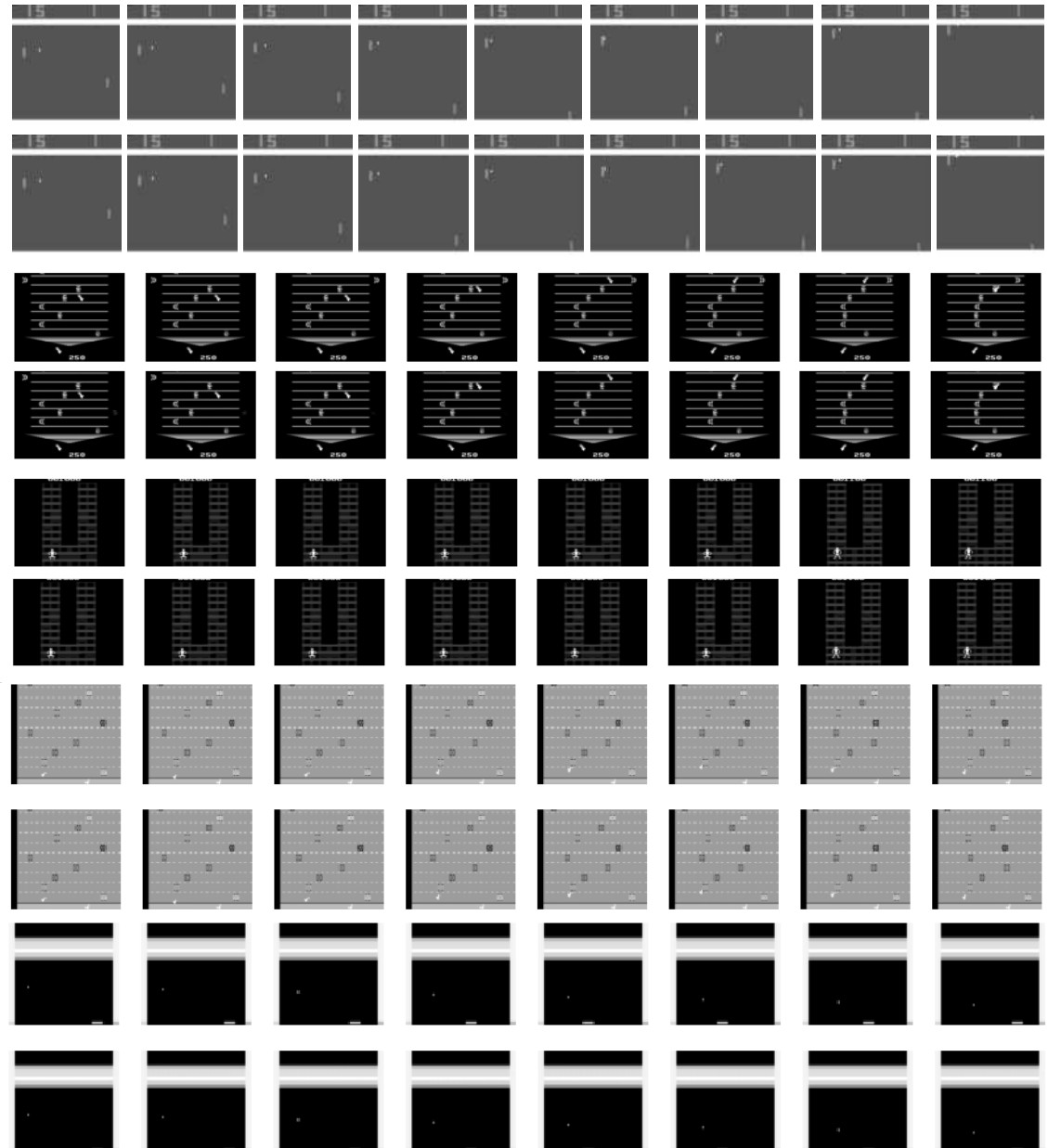

Figure 1: On the performance of the proposed GDM. Given four consecutive frames of Atari games, and a sequence of eight actions, GDM generates sequences of the future frames almost identical to the real frames. First row: A sequence of real frames. Second row: a corresponding sequence of generated frames

2016a). Furthermore, more general settings like partial monitoring games are theoretically tackled (Bartók et al., 2014) and minmax regret guarantees are provided.

While theoretical RL addresses variety of the trade-offs in the exploration-exploitation, this problem is still prominent in practical reinforcement learning research (Mnih et al., 2015; Abel et al., 2016; Azizzadenesheli et al., 2016b). On the empirical side, recent successes in video games has sparked a flurry of research interest. For example (Cuayáhuitl, 2016; Fatemi et al., 2016; Wen et al., 2016) investigate DRL for dialogue policy learning, with (Lipton et al., 2018) addressing the efficiency of exploration. To combat the sample complexity shortcoming, designing an efficient exploration strategy in DRL has emerged as an active research topic, e.g. optimism (Ostrovski et al., 2017) and Thompson Sampling (Osband et al., 2016; Lipton et al., 2018; Azizzadenesheli et al., 2018).

Minimizing the Bellman residual using Bootstraps of the $Q$-function has been the core of value based DRL methods (Mnih et al., 2015; Van Hasselt et al., 2016). Moreover, it has been extensively studied

that minimizing the Bellman residual provides a biased estimator of the value function (Antos et al., 2008; Thrun & Schwartz, 1993). In order to mitigate this bias, DQN proposes to update the target value less frequently than the rest of the model in order to mimic the Fitted-$Q$ update. This tweak might reduce the bias in the value estimator but significantly increases the sample complexity. On the other hand, Monte Carlo sampling strategies (Kearns et al., 2002; Kocsis & Szepesvári, 2006) have been proposed as efficient methods for planning, but suffer from high sample complexity in real world applications. Our results provide a deeper understanding of this bias in $Q$ and its relationship to model-based planning.

Despite GANs' capabilities at generating perceptually realistic images, they are difficult to train and often unstable, especially for non-stationary tasks like RL. In recent years, there has been significant progress in developing stable learning procedures. The Wasserstein GAN (W-GAN) (Arjovsky et al., 2017) uses the Wasserstein metric as a notion of distance between two distributions, while requires the discriminator to be from the set of bounded Lipschitz functions. In order to satisfied this boundedness, Gulrajani et al. (2017) proposed the improved W-GAN, which penalizes the discriminator's gradient although it still hard to train. Spectral normalization of discriminators (Miyato et al., 2018) has been studied recently, and has been shown empirically to converge more reliably. We leverage these advances in creating a stable learning procedure for the GDM.

Recently, conditional video prediction has emerged as a growing area of research. Previous work trains large models with L2 loss to predict long future trajectories of frames given actions (Oh et al., 2015). The quality of the generated frames is measured by training DQN on them. However, since these models struggle to produce high frequency details and cannot produce meaningful frames in stochastic environments, due to the L2 loss. We implemented this work and compared it against GDM, a much smaller architecture with discriminative loss. We observe that GDM requires significantly fewer iterations to converge to perceptually unidentifiable frames. We also observed significantly lower error for GDM when a $Q$ function is applied to generated frames from both models.

Finally, learned environment models, such as those used for conditional video prediction, are leveraged in Weber et al. (2017). A learned model encodes the generated trajectories into an abstract representation, which is used as an additional input to the policy model. They validate their methods on Sokoban, a small puzzle world, and show the capability of their model on multi-task learning in their miniPacman environment. Unlike GATS, Weber et al. (2017) does not use explicit planning and roll-out strategies. A similar approach to GATS is concurrently developed and empirically studied on Car Racing and VizDoom (David Ha, 2018). Further work employs transition models in order to perform rollouts in the encoded state representation (Oh et al., 2017), and demonstrate modest gains on Atari games (compared to DQN). A similar approach also has been studied in robotics (Wahlström et al., 2015). In contrast, we are able to learn model dynamics in the original state/pixel space.

GATS synthesizes this prior work into a flexible framework for studying model-based and model-free reinforcement learning with four basic building blocks: ($i$) value learning ($ii$) planning ($iii$) a reward predictor, and ($iv$) dynamics model. This freedom in the GATS framework allows for many different variations and adaptations for a given domain and problem, and thus provides many avenues for further exploration. For instance, for value learning ($i$), one can use Count-based methods (Bellemare et al., 2016). For planning ($ii$), one can use upper confidence bound tree search (UCT) (Kocsis & Szepesvári, 2006) or policy gradient methods (Kakade, 2002; Schulman et al., 2015). For the reward model ($iii$), if the reward has a continuous distribution, one can learn the mean reward using any regression model. Lastly, for model dynamics ($iv$), one can extend GDM or choose any other image generating model. Interestingly, this work can be extended to the $\lambda$-return setting, where a mix of $n$ steps are acquired. While GATS is an appealing and flexible RL paradigm, it suffers from high computation cost due to modeling in image space and due to MCTS. Potentially, we could mitigate this overhead with parallelization or distilled policy methods (Guo et al., 2014) by approximating with a smaller network.

## 7 DISCUSSION

**Discussion of negative results** In this section, we enumerate several hypotheses for why GATS under-performs DQN despite near-perfect modeling, and discuss several attempts to improve GATS based on these hypotheses. The following are shown in Table 1.

Table 1: Set of approaches explored to improve GATS performance

| Replay Buffer | Optimizer | Sampling strategies | Optimism |
|---|---|---|---|
| (i) Plain DQN (ii) Dyna-Q | (i) Learning rate (ii) Mini-batch size | (i) Leaf nodes (ii) Random samples from the tree (iii) Samples by following greedy Q (iv) Samples by following $\varepsilon$-greedy Q (v) Geometric distribution | (i)W-loss (ii) exp(W-loss) (iii) L1+L2+W-distance (iv) exp(L1+L2+\|W-distance\|) |

*Replay Buffer:* The agent's decision under GATS sometimes differs from that of the learned $Q$ model. Therefore, it is important that we allow the $Q$-learner to observe the outcome of important outcomes in the generated MCTS states. To address this problem, we tried storing the samples generated in tree search and use them to further train the $Q$-model. We studied two scenarios: (i) using plain DQN with no generated samples and (ii) using Dyna-Q to train the $Q$ function on the generated samples in MCTS. However, these techniques did not improve the performance of GATS.

*Optimizer:* Since the problem is slightly different from DQN, especially in the Dyna-Q setting with generated frames, we tried a variety of different learning rates and minibatch sizes to tune the $Q$-learner.

*Sampling strategy:* We considered variety of different ways to use the samples generated in tree search for learning in Dyna-Q. (i) Since we use the $Q$-function on the leaf nodes, we tried using the generated experience at the leaf nodes in the replay buffer. (ii) We randomly sampled additional generated experience from the tree to update the $Q$-learner. (iii) We choose the generated experience corresponding to the greedy action from the $Q$-learner on the generated tree by MCTS, which represents the trajectory we would have received following the greedy $Q$ algorithm. We hypothesized that if we trained $Q$ on its own decisions it would improve the learned $Q$-function. (iv) We also considered the case of following the $\varepsilon$-greedy policy induced by $Q$, rather than the greedy $Q$ itself. (v) Finally, since following GATS or $Q$ results in a bigger shift in the later part of tree, we used generated trajectories from the greedy and $\varepsilon$-greedy policies and stored experience which happened in the later part of tree with higher probability. Specifically, we tried a variety of different geometric distributions to see which one was most helpful.

*Optimism:* Optimism-based exploration strategy with the GDM. We observed that areas that of the state space that are novel to the GDM are often explored less, so the GDM has a higher absolute value of the Wasserstein distance for rarely visited state action pairs and a lower value of the Wasserstein distance for frequently seen state-action pairs. We added (i) the $W$-loss and also (ii) its exponent as a notion of extrinsic reward to encourage exploration. In (iii) and (iv) We did the same with a summation of different losses.

Despite this extensive and costly study on GATS, we were not able to show that GATS benefits besides a limited improvement in training speed on Pong.

**Hypothesis on negative results.** From these negative results, we propose a hypothesis for why tree search with short roll-outs, such as the GATS algorithm, might not boost performance even with perfect modeling with GDM and RP, despite reducing local bias. Consider the hypothetical situation described in Fig. 2(a) where a fish starts with an initialization of the $Q$ function such that the greedy action is represented by yellow arrows. If the fish follows DQN, it reaches the shark quickly and receives a negative reward. The agent learns the down action is not a good action. Now consider the GATS algorithm with the same $Q$-function and 2 step look-ahead, even with a true model simulator. In this case, when the agent reaches the step above the sharks, the MCTS roll-out informs the agent that there is a negative reward of going down. Thus, the agent chooses action right, following the GATS action (red action). This holds for all following states with the red arrow in them. GATS locally avoid bad states, but does this information globally propagate?

In this case, with $\varepsilon$-greedy exploration, the agent finally dies and gets a negative reward. This negative reward happens much later than for a DQN agent. Therefore, many more updates are required for the agent to learn that action down at the beginning of the game is not a good action. Moreover, this negative signal may even vanish due to the long update lengths. This shows how GATS roll-outs

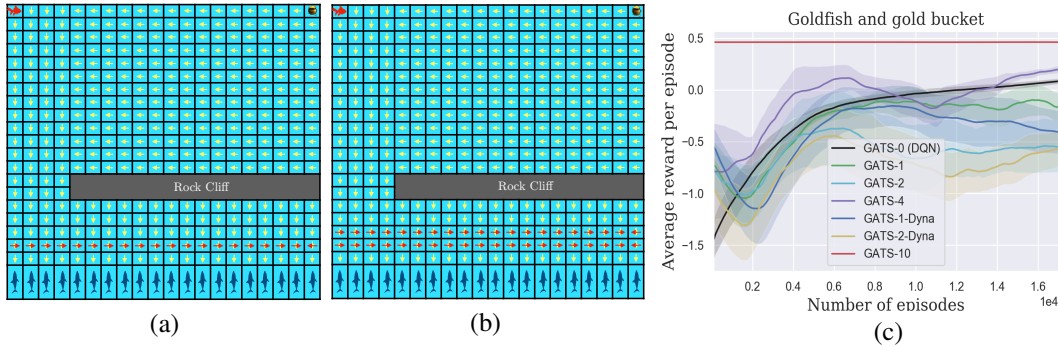

Figure 2: Goldfish looking for a Gold bucket. The Q function is initialized such that the yellow arrows represent the greedy action of each state. The red arrows are the actions suggested by following MCTS on the learned model. (a) Following GATS with depth two locally prevents the goldfish from hitting the sharks but slows down learning that action down is sub-optimal due to the delay in negative signal. (b) Even if the goldfish uses the prediction of the future event for further learning, it might just slightly mitigate the slow-down in the learning process, but the fundamental issue is still present, and the agent suffers from a slow-down in learning. (c) For a grid world of $10 \times 10$, GATS with depth of 10 (GATS-10) results in the highest return. Moreover, GATS with nonzero depth locally saves the agent from hitting the sharks, but in the long run it degrades the performance.

can locally help to avoid (or reach) catastrophic (or good) events, but it can also slow down global understanding.

As we suggested in our negative results, this problem can be solved by putting the generated experience of MCTS in the replay buffer and deploying Dyna-Q. However, Fig. 2(b) illustrates the situation where the GATS action in the third row before the sharks is "right". In this case, two-step roll-outs do not see the sharks, and thus Dyna-Q does not speed learning, all while requiring significantly more computation. In practice, especially with limited memory of the replay buffer and limited capacity of the function classes, it can be difficult to tune Dyna-Q to work well.

As the *Goldfish and gold bucket* experiment shows, deploying MCTS with Q-learning can have complex interactions. We showed in Proposition 1 that given a fixed estimated Q function, MCTS results in a better worst-case error in the Q estimation. However, this does not guarantee that learning the Q function while performing MCTS will not result in a worse estimated Q function. When this worse estimated Q-function is used in the leaf nodes with MCTS, it can result in worse performance, as indicated in our empirical results.

To test this hypotheses empirically in a controlled environment, we implemented the 10x10 version of *Goldfish and gold bucket* environment. We tested GATS with a depth of 0 (i.e., the plain DQN), 1, 2, 4 and GATS +Dyna-Q with the depth of 1 and 2. Figure 2(c) represents the per episode return of different algorithms. Each episode has a maximum length of 100 steps unless the agent either reaches the gold bucket (the reward of $+1$) or hit any of the sharks (the reward of $-1$). At each time step, the agent also suffers a cost of $0.05$ for not accomplishing the task, while the discount factor is $0.99$. We train the Q-network using DQN and a mini-batch of size $64$. We observe that GATS with the depth of 10 (the dimension of the grid) receives the highest return. Moreover, we observe that GATS with nonzero depth locally saves the agent to hit the sharks, since in the initial phase, GATS with non-zero depth has higher return than DQN. However, in the long run, GATS with short roll-outs (e.g. GATS-1 and GATS-2) degrades the performance, as seen in the later parts of the run. Furthermore, we see that the Dyna-Q approach also might fail in improving the performance in the naive case. We train GATS +Dyna-Q with both executed experiences and predicted ones in the oracle dynamics model. We observe that GATS +Dyna-Q does not provide much benefit over GATS. Without sophisticated sampling algorithms, GATS +Dyna-Q can cause the high-capacity network to overfit to the repeated samples.

For many complex applications, like hard Atari games, GATS may require significantly longer roll-out depths with sophisticated Dyna-Q in order to perform well, which was computationally in-feasible for this study. However, the insights in designing near-perfect modeling algorithms and the extensive study of GATS, highlight several key considerations and provide an important framework to effectively design algorithms combining model-based and model-free reinforcement learning.

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

## APPENDIX

## A PROOF OF PROPOSITION 1

Let's restate the estimated returns with the learned model as follows;

$$\mathbb{E}_{\pi_r,\text{GDM,RP}}\left[\sum_{h=0}^{H-1}\gamma^h\widehat{r}_h+\gamma^H\max_a\widehat{Q}(x_H,a)\Big|x\right] := \sum_{x_i,a_i,\forall i\in[1,..,H]}\widehat{T}(x_1|x,a_1)\pi_r(a_1|x)\prod_{j=2}^{H}\widehat{T}(x_j|x_{j-1},a_j)\pi_r(a_j|x_{j-1})$$

$$\left(\widehat{r}(x,a_1)+\sum_{j=2}^{H}\gamma^{j-1}\widehat{r}(x_{j-1},a_j)+\gamma^H\max_a\widehat{Q}(x_H,a)\right)$$

Now consider the following lemma;

**Lemma 1 (Deviation in Q-function)** *Define $e_Q$ as the uniform bound on error in the estimation of the Q function, such that $|Q(x,a)-\widehat{Q}(x,a)|\le e_Q$ , $\forall x,a$. Then;*

$$\left|\max_a\widehat{Q}(x,a)-\max_aQ(x,a)\right|\le e_Q$$

**Proof 1** *For a given state $x$, define $\widetilde{a}_1(x):=\arg\max_a\widehat{Q}(x,a)$ and $\widetilde{a}_2(x):=\arg\max_aQ(x,a)$. Then;*

$$\widehat{Q}(x,\widetilde{a}_1(x))-Q(x,\widetilde{a}_2(x))=\widehat{Q}(x,\widetilde{a}_1(x))-Q(x,\widetilde{a}_1(x))+Q(x,\widetilde{a}_1(x))-Q(x,\widetilde{a}_2(x))$$
$$\le\widehat{Q}(x,\widetilde{a}_1(x))-Q(x,\widetilde{a}_1(x))\le e_Q$$

*since $Q(x,\widetilde{a}_1(x))-Q(x,\widetilde{a}_2(x))\le 0$.*

*With similar argument, we have;*

$$\widehat{Q}(x,\widetilde{a}_1(x))-Q(x,\widetilde{a}_2(x))=\widehat{Q}(x,\widetilde{a}_1(x))-\widehat{Q}(x,\widetilde{a}_2(x))+\widehat{Q}(x_H,\widetilde{a}_2(x))-Q(x,\widetilde{a}_2(x))$$
$$\ge\widehat{Q}(x,\widetilde{a}_2(x))-Q(x,\widetilde{a}_2(x))\ge -e_Q$$

*since $\widehat{Q}(x,\widetilde{a}_1(x))-\widehat{Q}(x,\widetilde{a}_2(x))\ge 0$. Therefore;*

$$-e_Q\le\widehat{Q}(x,\widetilde{a}_1(x))-Q(x,\widetilde{a}_2(x))\le e_Q$$

*resulting in Lemma 1.*

In the remaining proof, we repeatedly apply the addition and subtraction technique to upper bound the error. To show how we apply this technique, we illustrate how we derive the error terms $T(x_1|x,a_1)-\widehat{T}(x_1|x,a_1)$ and $r(x,a_1)-\widehat{r}(x,a_1)$ for the first time step in detail.

Let us restate the objective that we desire to upper bound:

$$\left|\mathbb{E}_{\pi_r}\left[\sum_{h=0}^{H-1}\gamma^hr_h+\gamma^H\max_aQ(x_H,a)\Big|x\right]-\mathbb{E}_{\pi_r,\text{GDM,RP}}\left[\sum_{h=0}^{H}\gamma^h\widehat{r}_h+\gamma^H\max_a\widehat{Q}(x_H,a)\Big|x\right]\right|$$

$$=\left|\sum_{x_i,a_i,\forall i\in[1,..,H]}T(x_1|x,a_1)\pi_r(a_1|x)\prod_{j=2}^{H}T(x_j|x_{j-1},a_j)\pi_r(a_j|x_{j-1})\left(r(x,a_1)+\sum_{j=2}^{H}\gamma^{j-1}r(x_{j-1},a_j)+\gamma^H\max_aQ(x_H,a)\right)\right.$$

$$\left.-\sum_{x_i,a_i,\forall i\in[1,..,H]}\widehat{T}(x_1|x,a_1)\pi_r(a_1|x)\prod_{j=2}^{H}\widehat{T}(x_j|x_{j-1},a_j)\pi_r(a_j|x_{j-1})\left(\widehat{r}(x,a_1)+\sum_{j=2}^{H}\gamma^{j-1}\widehat{r}(x_{j-1},a_j)+\gamma^H\max_a\widehat{Q}(x_H,a)\right)\right|$$

$$\tag{8}$$

Then, we add and subtract the following term.

$$\sum_{x_i,a_i,\forall i \in [1,..,H]} \widehat{T}(x_1|x,a_1)\pi_r(a_1|x) \prod_{j=2}^{H} T(x_j|x_{j-1},a_j)\pi_r(a_j|x_{j-1}) \left( r(x,a_1) + \sum_{j=2}^{H}\gamma^{j-1}r(x_{j-1},a_j) + \gamma^H \max_a Q(x_H,a)\right)$$

Notice this term differs from the first term of Eq. 8 just in the transition kernel of the first time step, i.e., $T(x_1|x,a_1) \to \widehat{T}(x_1|x,a_1)$. Thus, we have;

$$\left| \mathbb{E}_{\pi_r}\left[ \sum_{h=0}^{H-1}\gamma^h r_h + \gamma^H \max_a Q(x_H,a)\Big|x\right] - \mathbb{E}_{\pi_r,\text{GDM,RP}}\left[ \sum_{h=0}^{H}\gamma^h \widehat{r}_h + \gamma^H \max_a \widehat{Q}(x_H,a)\Big|x\right] \right|$$

$$= \Bigg| \sum_{x_i,a_i,\forall i \in [1,..,H]} T(x_1|x,a_1)\pi_r(a_1|x) \prod_{j=2}^{H} T(x_j|x_{j-1},a_j)\pi_r(a_j|x_{j-1}) \left( r(x,a_1) + \sum_{j=2}^{H}\gamma^{j-1}r(x_{j-1},a_j) + \gamma^H \max_a Q(x_H,a)\right)$$

$$- \sum_{x_i,a_i,\forall i \in [1,..,H]} \widehat{T}(x_1|x,a_1)\pi_r(a_1|x) \prod_{j=2}^{H} T(x_j|x_{j-1},a_j)\pi_r(a_j|x_{j-1}) \left( r(x,a_1) + \sum_{j=2}^{H}\gamma^{j-1}r(x_{j-1},a_j) + \gamma^H \max_a Q(x_H,a)\right)$$

$$+ \sum_{x_i,a_i,\forall i \in [1,..,H]} \widehat{T}(x_1|x,a_1)\pi_r(a_1|x) \prod_{j=2}^{H} T(x_j|x_{j-1},a_j)\pi_r(a_j|x_{j-1}) \left( r(x,a_1) + \sum_{j=2}^{H}\gamma^{j-1}r(x_{j-1},a_j) + \gamma^H \max_a Q(x_H,a)\right)$$

$$- \sum_{x_i,a_i,\forall i \in [1,..,H]} \widehat{T}(x_1|x,a_1)\pi_r(a_1|x) \prod_{j=2}^{H} \widehat{T}(x_j|x_{j-1},a_j)\pi_r(a_j|x_{j-1}) \left( \widehat{r}(x,a_1) + \sum_{j=2}^{H}\gamma^{j-1}\widehat{r}(x_{j-1},a_j) + \gamma^H \max_a \widehat{Q}(x_H,a)\right)\Bigg|$$

$$(9)$$

We derive the error term $T(x_1|x,a_1) - \widehat{T}(x_1|x,a_1)$ from the first two terms of Eq. 9. Notice that all the parameters of the first two terms are the same except the transition kernel for the first state. We can thus refactor the first two terms of Eq. 9 as:

$$\sum_{x_i,a_i,\forall i \in [1,..,H]} \left( T(x_1|x,a_1) - \widehat{T}(x_1|x,a_1)\right)\pi_r(a_1|x) \prod_{j=2}^{H} T(x_j|x_{j-1},a_j)\pi_r(a_j|x_{j-1})$$

$$\left( r(x,a_1) + \sum_{j=2}^{H}\gamma^{j-1}r(x_{j-1},a_j) + \gamma^H \max_a Q(x_H,a)\right)$$

We then expand the third and fourth terms of Eq. 9 to derive the error term $r(x,a_1) - \widehat{r}(x,a_1)$ and a remainder. To do this, we add and subtract the following term which is the same as the third term in Eq. 9 except it differs in the reward of the first time step:

$$\sum_{x_i,a_i,\forall i \in [1,..,H]} \widehat{T}(x_1|x,a_1)\pi_r(a_1|x) \prod_{j=2}^{H} T(x_j|x_{j-1},a_j)\pi_r(a_j|x_{j-1}) \left( \widehat{r}(x,a_1) + \sum_{j=2}^{H}\gamma^{j-1}r(x_{j-1},a_j) + \gamma^H \max_a Q(x_H,a)\right)$$

We can thus express the third and fourth terms of Eq. 9 along with the addition and subtraction terms as:

$$
\sum_{x_i, a_i, \forall i \in [1,.,H]} \widehat{T}(x_1|x, a_1) \pi_r(a_1|x) \prod_{j=2}^{H} T(x_j|x_{j-1}, a_j) \pi_r(a_j|x_{j-1}) \left( r(x, a_1) + \sum_{j=2}^{H} \gamma^{j-1} r(x_{j-1}, a_j) + \gamma^H \max_a Q(x_H, a) \right)
$$

$$
- \sum_{x_i, a_i, \forall i \in [1,.,H]} \widehat{T}(x_1|x, a_1) \pi_r(a_1|x) \prod_{j=2}^{H} T(x_j|x_{j-1}, a_j) \pi_r(a_j|x_{j-1}) \left( \widehat{r}(x, a_1) + \sum_{j=2}^{H} \gamma^{j-1} r(x_{j-1}, a_j) + \gamma^H \max_a Q(x_H, a) \right)
$$

$$
+ \sum_{x_i, a_i, \forall i \in [1,.,H]} \widehat{T}(x_1|x, a_1) \pi_r(a_1|x) \prod_{j=2}^{H} T(x_j|x_{j-1}, a_j) \pi_r(a_j|x_{j-1}) \left( \widehat{r}(x, a_1) + \sum_{j=2}^{H} \gamma^{j-1} r(x_{j-1}, a_j) + \gamma^H \max_a Q(x_H, a) \right)
$$

$$
- \sum_{x_i, a_i, \forall i \in [1,.,H]} \widehat{T}(x_1|x, a_1) \pi_r(a_1|x) \prod_{j=2}^{H} \widehat{T}(x_j|x_{j-1}, a_j) \pi_r(a_j|x_{j-1}) \left( \widehat{r}(x, a_1) + \sum_{j=2}^{H} \gamma^{j-1} \widehat{r}(x_{j-1}, a_j) + \gamma^H \max_a \widehat{Q}(x_H, a) \right)
$$

$$(10)$$

Notice that the first two terms in Eq. 10 are the same except in in the first reward term, from which we derive the error term $r(x, a_1) - \widehat{r}(x, a_1)$. We refactor the first two terms in Eq. 10 as:

$$
\sum_{x_i, a_i, \forall i \in [1,.,H]} (r(x, a_1) - \widehat{r}(x, a_1)) \widehat{T}(x_1|x, a_1) \pi_r(a_1|x) \prod_{j=2}^{H} T(x_j|x_{j-1}, a_j) \pi_r(a_j|x_{j-1})
$$

Finally, we have the remaining last two terms of Eq. 10.

$$
\sum_{x_i, a_i, \forall i \in [1,.,H]} \widehat{T}(x_1|x, a_1) \pi_r(a_1|x) \prod_{j=2}^{H} T(x_j|x_{j-1}, a_j) \pi_r(a_j|x_{j-1}) \left( \widehat{r}(x, a_1) + \sum_{j=2}^{H} \gamma^{j-1} r(x_{j-1}, a_j) + \gamma^H \max_a Q(x_H, a) \right)
$$

$$
- \sum_{x_i, a_i, \forall i \in [1,.,H]} \widehat{T}(x_1|x, a_1) \pi_r(a_1|x) \prod_{j=2}^{H} \widehat{T}(x_j|x_{j-1}, a_j) \pi_r(a_j|x_{j-1}) \left( \widehat{r}(x, a_1) + \sum_{j=2}^{H} \gamma^{j-1} \widehat{r}(x_{j-1}, a_j) + \gamma^H \max_a \widehat{Q}(x_H, a) \right)
$$

We repeatedly expand this remainder for the following time steps using the same steps as described above to derive the full bound. Following this procedure, we have:

$$\left| \mathbb{E}_{\pi_r} \left[ \sum_{h=0}^{H-1} \gamma^h r_h + \gamma^H \max_a Q(x_H, a) \Big| x \right] - \mathbb{E}_{\pi_r, \text{GDM,RP}} \left[ \sum_{h=0}^{H} \gamma^h \widehat{r}_h + \gamma^H \max_a \widehat{Q}(x_H, a) \Big| x \right] \right|$$

$$\leq \sum_{x_i, a_i, \forall i \in [H]} \left| T(x_1|x, a_1) - \widehat{T}(x_1|x, a_1) \right| \pi_r(a_1|x)$$

$$\left( r(x, a_1) + \sum_{j=2}^{H} \gamma^{j-1} r(x_{j-1}, a_j) + \gamma^H \max_a Q(x_H, a) \right) \prod_{j=2}^{H} T(x_j|x_{j-1}, a_j) \pi_r(a_j|x_{j-1})$$

$$+ \sum_{x_i, a_i, \forall i \in [H]} \widehat{T}(x_1|x, a_1) \pi_r(a_1|x) \left( |r(x, a_1) - \widehat{r}(x, a_1)| \right) \prod_{j=2}^{H} T(x_j|x_{j-1}, a_j) \pi_r(a_j|x_{j-1})$$

$$+ \sum_{j=2}^{H} \sum_{x_h, a_h, \forall i \in [H]} \widehat{T}(x_1|x, a_1) \pi_r(a_1|x) \left| T(x_j|x_{j-1}, a_j) - \widehat{T}(x_j|x_{j-1}, a_j) \right|$$

$$\left( \sum_{h=j+1}^{H} \gamma^{h-1} r(x_{h-1}, a_h) + \gamma^H \max_a Q(x_H, a) \right)$$

$$\prod_{h=2}^{j-1} \widehat{T}(x_h|x_{h-1}, a_h) \pi_r(a_h|x_{h-1}) \prod_{h=j+1}^{H} T(x_h|x_{h-1}, a_h) \pi_r(a_h|x_{h-1})$$

$$+ \sum_{j=2}^{H} \gamma^{j-1} \sum_{x_h, a_h, \forall i \in [H]} \widehat{T}(x_1|x, a_1) \pi_r(a_1|x) \left( \left| r(x_{j-1}, a_j) - \widehat{r}(x_{j-1}, a_j) \right| \right)$$

$$\prod_{h=2}^{j} \widehat{T}(x_h|x_{h-1}, a_h) \pi_r(a_h|x_{h-1}) \prod_{h=j+1}^{H} T(x_h|x_{h-1}, a_h) \pi_r(a_h|x_{h-1})$$

$$+ \sum_{x_h, a_h, \forall i \in [H]} \widehat{T}(x_1|x, a_1) \pi_r(a_1|x) \prod_{h=2}^{H} \widehat{T}(x_h|x_{h-1}, a_h) \pi_r(a_h|x_{h-1}) \gamma^H \left| \max_a Q(x_H, a) - \max_a \widehat{Q}(x_H, a)) \right|$$

As a result,

$$\left| \mathbb{E}_{\pi_r} \left[ \sum_{h=0}^{H-1} \gamma^h r_h + \gamma^H \max_a Q(x_H, a) \Big| x \right] - \mathbb{E}_{\pi_r, \text{GDM,RP}} \left[ \sum_{h=0}^{H-1} \gamma^h \widehat{r}_h + \gamma^H \max_a \widehat{Q}(x_H, a) \Big| x \right] \right|$$

$$\leq \sum_{i=1}^{H} \gamma^{i-1} \frac{1 - \gamma^{H+1-i}}{1 - \gamma} e_T + \sum_{i=1}^{H} \gamma^{i-1} e_R + \gamma^H e_Q$$

$$\leq \frac{1 - \gamma^H + H\gamma^H(1-\gamma)}{(1-\gamma)^2} e_T + \frac{1-\gamma^H}{1-\gamma} e_R + \gamma^H e_Q$$

Therefore, the proposition follows.

## B    BIAS-VARIANCE OF $Q_\theta$

To observe the existing bias and variance in $Q_\theta$, we run solely DQN on the game Pong, for $20M$ frame steps. Fig. 3 shows 4 consecutive frames where the agent receives a negative score and Table. 3 shows the estimated Q values by DQN for these steps. As we observe in Fig. 3 and Table. 3, at the time step $t$, the estimated Q value of all the actions are almost the same. The agent takes the *down* action and the environment goes to the next state $t + 1$. The second row of Table. 3 expresses the Q value of the actions at this new state. Since this transition does not carry any reward and the discount factor is close to 1, ($\gamma = 0.99$) we expect the max Q values at time step $t + 1$ to be close the $Q$ values



| | Action space | | |
|---|---|---|---|
| Steps | stay | up | down |
| $t$ | 4.3918 | 4.3220 | 4.3933 |
| $t+1$ | 2.7985 | 2.8371 | 2.7921 |
| $t+2$ | 2.8089 | 2.8382 | 2.8137 |
| $t+3$ | 3.8725 | 3.8795 | 3.8690 |

Figure 3: The sequence of four consecutive decision states, and corresponding learned $Q$-function by DQN at $t, t+1, t+2, t+3$ from left to right, where the agent loses the point. At time step $t$, the optimal action is *up* but the $Q$-value of going up is lower than other actions. More significantly, even though the agent chooses action *down* and goes down, the Q value of action down at time step $t$ is considerably far from the maximum Q value of the next state at time step $t+1$.

of action *down*, but it is very different. Moreover, in Fig. 4 and Table. 4 we investigate the case that the agent catches the ball. The ball is going to the right and agent needs to catch it. At time step $t$, the paddle is not on the direction of ball velocity, and as shown in Table. 4, the optimal action is *down*. But a closer look at the estimated Q value of action *up* reveals that the Q value for both action *up* is unreasonably close, when it could lead to losing the point. Lastly, we studied the existing errors

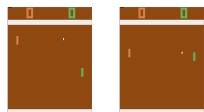

| | Action space | | |
|---|---|---|---|
| Steps | stay | up | down |
| $t$ | 1.5546 | 4.5181 | 4.5214 |

Figure 4: States at $t-1 \rightarrow t$ and the corresponding Q function learned through DQN at time $t$. Action *up* is sub-optimal but has high value and considerably close to action *down*. While action *down* and *stay* show have more similar values than *up* and *down*

in the estimation of the $Q$ function using DQN. In Table.3, if the agent could roll-out even one step before making a decision, it could observe negative consequence of action *down*. The positive effect of the roll-out is more significant in earlier stages of $Q$ learning, where the $Q$ estimation is more off.

## C    GATS ON PONG

We run GATS with $1, 2, 3$, and $4$ steps lookahead ($GATS1, GATS2, GATS3, GATS4$) and show its performance improvement over DQN in  Fig. 5 *(left)*.  Fig. 5 *(right)* shows the RP prediction accuracy. We observe that when the transition phase occurs at decision step 1M, the RP model misclassifies the positive rewards. But the RP rapidly adapts to this shift and reduces the classification error to less than 2 errors per episode.

As DRL methods are data hungry, we can re-use the data to efficiently learn the model dynamics. Fig. 1 shows how accurate the GDM can generate next 9 frames just conditioning on the first frame and the trajectory of actions. This trajectory is generated at decision step $100k$. Moreover, we extend our study to the case where we change the model dynamics by changing the game mode. In this case, by going from default mode to alternate mode in pong, the opponent paddle gets halved in size. We expected that in this case, where the game became easier, the DDQN agent would preserve its performance but surprisingly it gave us the most negative score possible, i.e -21 and broke. Therefore, we start fine tuning DDQN and took 3M time step (12M frame) to master the game again. It is worth noting that it takes DDQN 5M time step (20M frame) to master from scratch. While DDQN shows a vulnerable and undesirably breakable behaviour to this scenario, GDM and RP thanks to their detailed design, adapt to the new model dynamics in 3k samples, which is amazingly smaller (see detail in F)

In addition to GATS on DQN, we also study two other set of experiments on DDQN. Since Fig. 5 shows that the deeper roll-outs beyond one step do not provide much additional benefit for Pong, we focus on one-step roll-outs for the next two experiments. In the first experiment, we equip GATS +DDQN with the mentioned Wasserstein-optimism approach, and compare it with DDQN and plain GATS +DDQN, which both use $\varepsilon$-greedy based approaches for exploration. In Fig. 6*left*,  we observe that this optimism heuristic is helpful for better exploration.

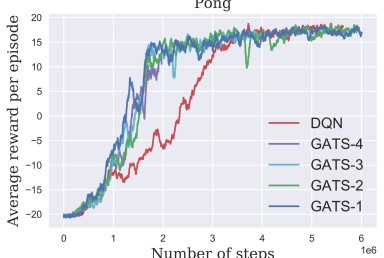 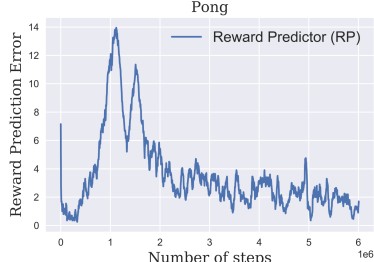

Figure 5: *left*:GATS learns a better policy faster than plain DQN (2 times faster). GATS $k$ denotes GATS of depth $k$. *right*: Accuracy of RP. The $Y$ axis shows the number of mistakes per episode and each episode has average length of $2k$, so the acc is almost always around $99.8\%$. This accuracy is consistent among runs and different lookahead lengths.

In the second experiment, we investigate the effect of prioritizing training samples for the GDM, fresher samples are more probable to be chosen, which we do in all experiments reported in Fig. 6*left*. We study the case where the input samples to GDM are instead chosen uniformly at random from the replay buffer in Fig. 6*right*. In this case the GATS learns a better policy faster at the beginning of the game, but the performance stays behind DDQN, due to the shift in the state distribution. It is worth mentioning that for optimism based exploration, there is no $\varepsilon$-greedy, which is why it gets close to the maximum score of 21. We tested DDQN and GATS-DDQN with $\varepsilon = 0$, and they also perform close to 21. We further extend the study of GDM to more games 1 and observed same robust behaviour as Pong. We also tried to apply GATS to more games, but were not able to extend it due to mainly its high computation cost. We tried different strategies of storing samples generated by MCTS, e.g. random generated experience, trajectory followed by Q on the tree, storing just leaf nodes, max leaf, also variety of different distributions, e.g. geometric distributing, but again due the height cost of hyper parameter tuning we were not successful to come up with a setting that GATS works for other games

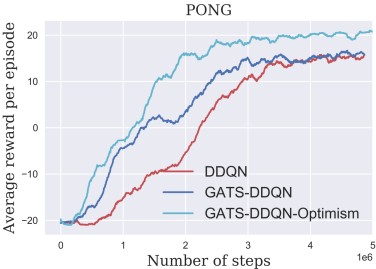 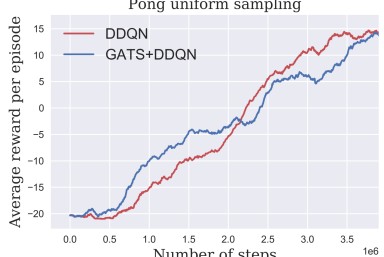

Figure 6: *left*:The optimism approach for GATS improves the sample complexity and learns a better policy faster. *right*: Sampling the replay buffer uniformly at random to train GDM, makes GDM slow to adapt to novel parts of state space.

## D   ASTERIX AND BREAKOUT NEGATIVE RESULTS

We include the results for GATS with 1 step look-ahead (GATS-1) and compare its performance to DDQN as an example for the negative results we obtained with short roll-outs with the GATS algorithm. While apply the same hyper parameters we tuned for pong, for Asterix results in performance slightly above random policy, we re-do the hyper-parameter tuning specifically for this game again and Fig. 7 is the best performance we achieved.

This illustrates the challenges of learning strong global policies with short roll-outs even with near-perfect modeling.

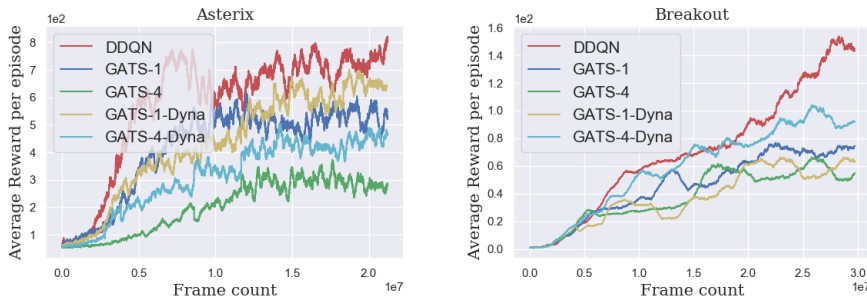

Figure 7: The GATS algorithm on Asterix and Breakout with 1 and 4 step look-ahead, compared to the DDQN baseline.

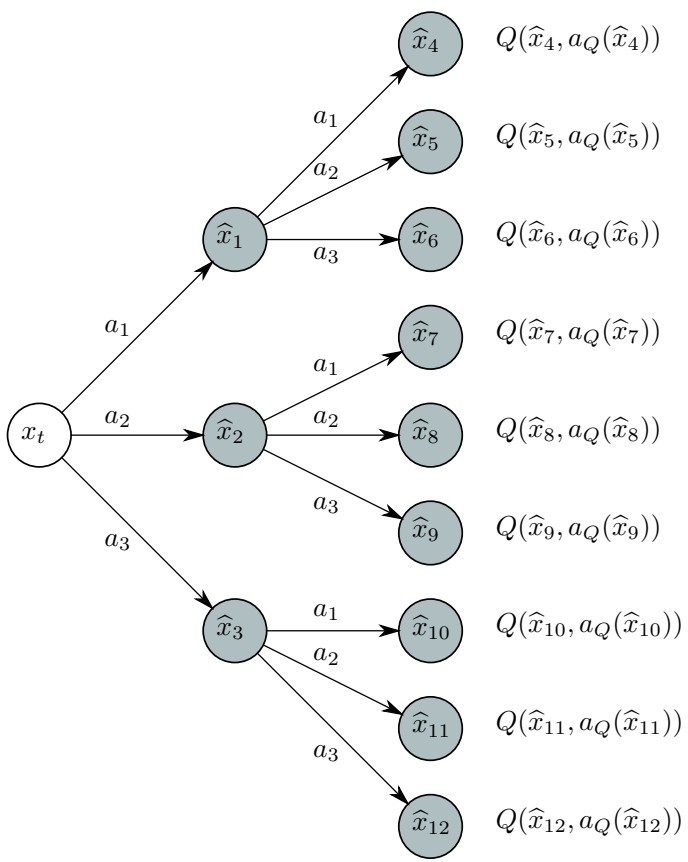

Figure 8: Roll-out of depth two starting from state $x_t$. Here $\widehat{x}$'s are the generated states by GDM. $Q(x, a(x))$ denotes the predicted value of state $x$ choosing the greedy action $a_Q(x) := \arg\max_{a' \in \mathcal{A}} Q(x, a')$.

## E GDM ARCHITECTURE AND PARAMETERS

The GDM model consists of seven convolution and also seven deconvolution layers. Each convolution layer is followed by Batch Normalization layers and the leaky RLU activation function with negative slope of $-0.2$. Also each deconvolution layer is followed by a Batch Normalization layer and the RELU activation instead of leaky RELU. The encoder part of the network uses channel dimensions of $32, 32, 64, 128, 256, 512, 512$ and kernel sizes of $4, 4, 4, 4, 2, 2, 2$. The reverse is true for the decoder

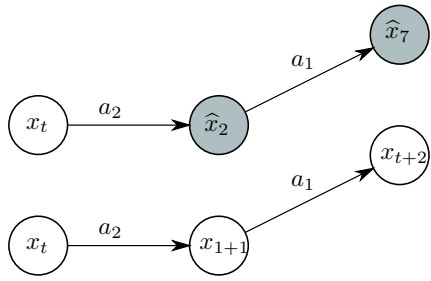

Figure 9: Training GAN and $Q_{\theta'}$ using the longer trajectory of experiences

part. We concatenate the bottleneck and next 5 deconvolution layers with a random Gaussian noise of dimension 100, the action sequence, and also the corresponding layer in the encoder. The last layer of decoder is not concatenated. Fig. 10. For the discriminator, instead of convolution, we use SN-convolution (Miyato et al., 2018) which ensures the Lipschitz constant of the discriminator is below 1. The discriminator consists of four SN-convolution layers followed by Batch Normalization layers and a leaky RELU activation with negative slope of $-0.2$. The number of channels increase as $64, 128, 256, 16$ with kernel size of $8, 4, 4, 3$, which is followed by two fully connected layers of size 400 and 18 where their inputs are concatenated with the action sequence. The output is a single number without any non-linearity. The action sequence uses one hot encoding representation.

We train the generator using Adam optimizer with weight decay of $0.001$, learning rate of $0.0001$ and also $beta1, beta2 = 0.5, 0.999$. For the discriminator, we use SGD optimizer with smaller learning rate of $0.00001$, momentum of $0.9$, and weight decay of $0.1$. Given the fact that we use Wasserstein metric for GDM training, the followings are the generator and discriminator gradient updates: for a given set of 5 frames and a action, sampled from the replay buffer, $(f_1, f_2, f_3, f_4, a_4, f_5)$ and a random Gaussian vector $z$:

Discriminator update:

$$\nabla_{\theta_D}\left[\frac{1}{m}\sum_{i=1}^{m}D_{\theta_D}(f_5, f_4, f_3, f_2, a_4) - \frac{1}{m}\sum_{i=1}^{m}D_{\theta_D}(G_{\theta_G}(f_4, f_3, f_2, f_1, a_4), f_4, f_3, f_2, a_4, z)\right]$$

Generator update:

$$\nabla_{\theta_G}\left[-\frac{1}{m}\sum_{i=1}^{m}D_{\theta_D}(G_{\theta_G}(f_4, f_3, f_2, f_1, a_4, z), f_4, f_3, f_2, a_4)\right]$$

where $\theta^{\mathrm{GDM}} = \{\theta_G, \theta_D\}$ are the generator parameters and discriminator parameters. In order to improve the quality of the generated frames, it is common to also add a class of multiple losses and capture different frequency aspects of the frames Isola et al. (2017); Oh et al. (2015). Therefore, we also add $10 * L1 + 90 * L2$ loss to the GAN loss in order to improve the training process. It is worth noting twh these losses are defined on the frames with pixel values in $[-1, 1]$, therefore they are small but still able to help speed up the the learning. In order to be able to roll-out for a longer and preserve the GDM quality, we also train the generator using self generated samples, i.e. given the sequence $(f_1, f_2, f_3, f_4, a_4, f_5, a_5, f_6, a_6, f_7, a_7, f_8)$, we also train the generator and discriminator on the generated samples of generator condition on its own generated samples for depth of three. This allows us to roll out for longer horizon of more than 10 and still preserve the GDM accuracy.

**Q function on generated frames** Ideally, if the GDM model is perfect at generating frames i.e. the space generated frames is, pixel by pixel, the same as the real frames, for the leaf nodes $x_H$, we can use $\max_a Q(x_H, a; \theta)$, learned by the DQN model on real frames, in order to assign values to the leaf nodes. But in practice, instead of $x_H$, we have access to $\widehat{x}_H$, a generated state twh perceptually is similar to $x_H$ (Fig. 1), but from the perspective of $Q_\theta$, they might not be similar over the course of training of $Q_\theta$. In order to compensate for this error, we train another $Q$-network, parameterized with $\theta'$, in order to provide the similar Q-value as $Q_\theta$ for generated frames. To train $Q_{\theta'}$, we minimize the $L2$ norm between $Q_{\theta'}$ and $Q_\theta$ for a given GAN sample state and trajectory

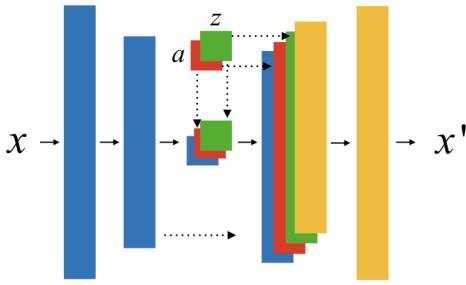

Figure 10: The GDM generator is an encoder-decoder architecture with skip-connections between mirrored layers, with action and Gaussian noise concatenated in the bottleneck and decoder layers.

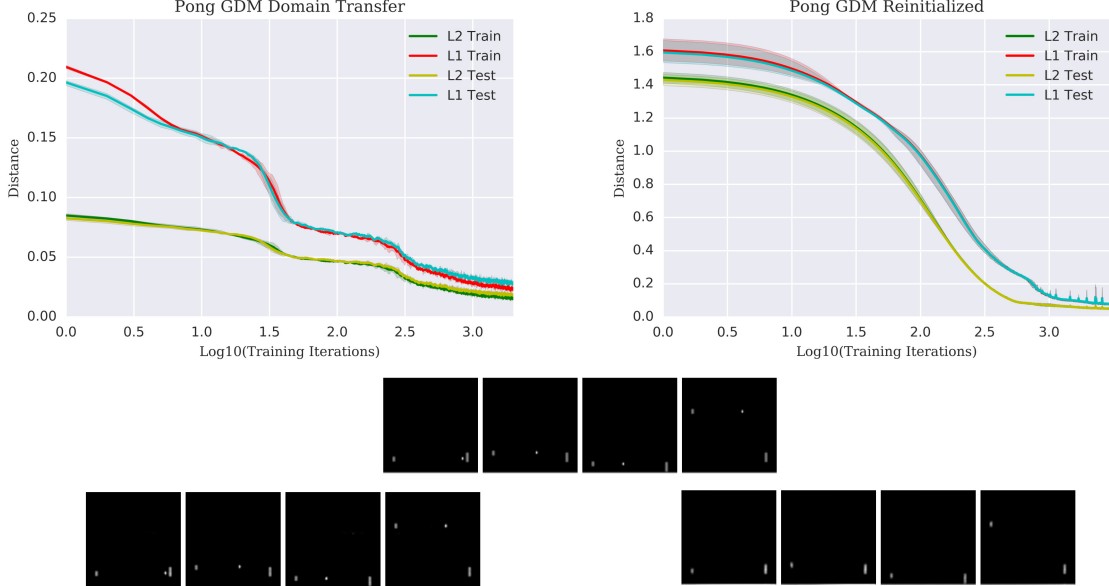

Figure 11: Training and evaluating domain transfer for GDM on new game dynamics for Pong (Mode 1, Difficulty 1). GDM domain transfer from Pong (Mode 0, Difficulty 0) on *left* and GDM from re-initialized parameters on *right*. L1 and L2 loss curves displayed *top*. Ground truth next frames displayed *middle* with predicted next frames displayed *bottom*.

Fig. 9. For this minimization, we use Adam with learning rate of $0.0001$, no weight decay, and $beta1, beta2 = 0.5, 0.999$. We experimented with weight decay and adding $L1$ loss, but we find these optimizations degrade the performance of the network. We tracked the difference between $Q_\theta(\widehat{x}) - Q_\theta(x)$ and $Q_{\theta'}(\widehat{x}) - Q_\theta(x)$ and observed twh both of these quantities are negligible. We ran GATS without the $Q\theta'$, with just $Q_\theta$, and observed only slightly worse performance.

## F GDM DOMAIN ADAPTATION.

We evaluate the GDM's ability to perform domain adaptation using the environment mode and difficulty settings in the latest Arcade Learning Environment (Machado et al., 2017). We first fully train GDM and DDQN on Pong with Difficulty 0 and Mode 0. We then sample 10,000 frames for training the GDM on Pong with Difficulty 1 and Mode 1, which has a smaller paddle and different game dynamics. We also collect 10,000 additional frames for testing the GDM. We train GDM using transferred weights and reinitialized weights on the new environment samples and observe the L1 and L2 loss on training and test samples over approximately 3,000 training iterations, and we observe

twh they decrease together without significant over-fitting in Fig. 11. To qualitatively evaluate these frames, we plot the next frame predictions of four test images in Fig. 11. We observe twh training GDM from scratch converges to a similarly low L1 and L2 loss quickly, but it fails to capture the game dynamics of the ball. This indicates the L1 and L2 loss are bad measurements of a model's ability to capture game dynamics. GDM is very efficient at transfer. It quickly learns the new model dynamics and is able to generalize to new test states with an order of magnitude fewer samples than the $Q$-learner.

## G    GDM TREE ROLLOUTS

Finally, we evaluate the ability of the GDM to generate different future trajectories from an initial state. We sample an initial test state and random action sequences of length 5. We then unroll the GDM for 5 steps from the initial state. We visualize the different rollout trajectories in Figs. 1213.

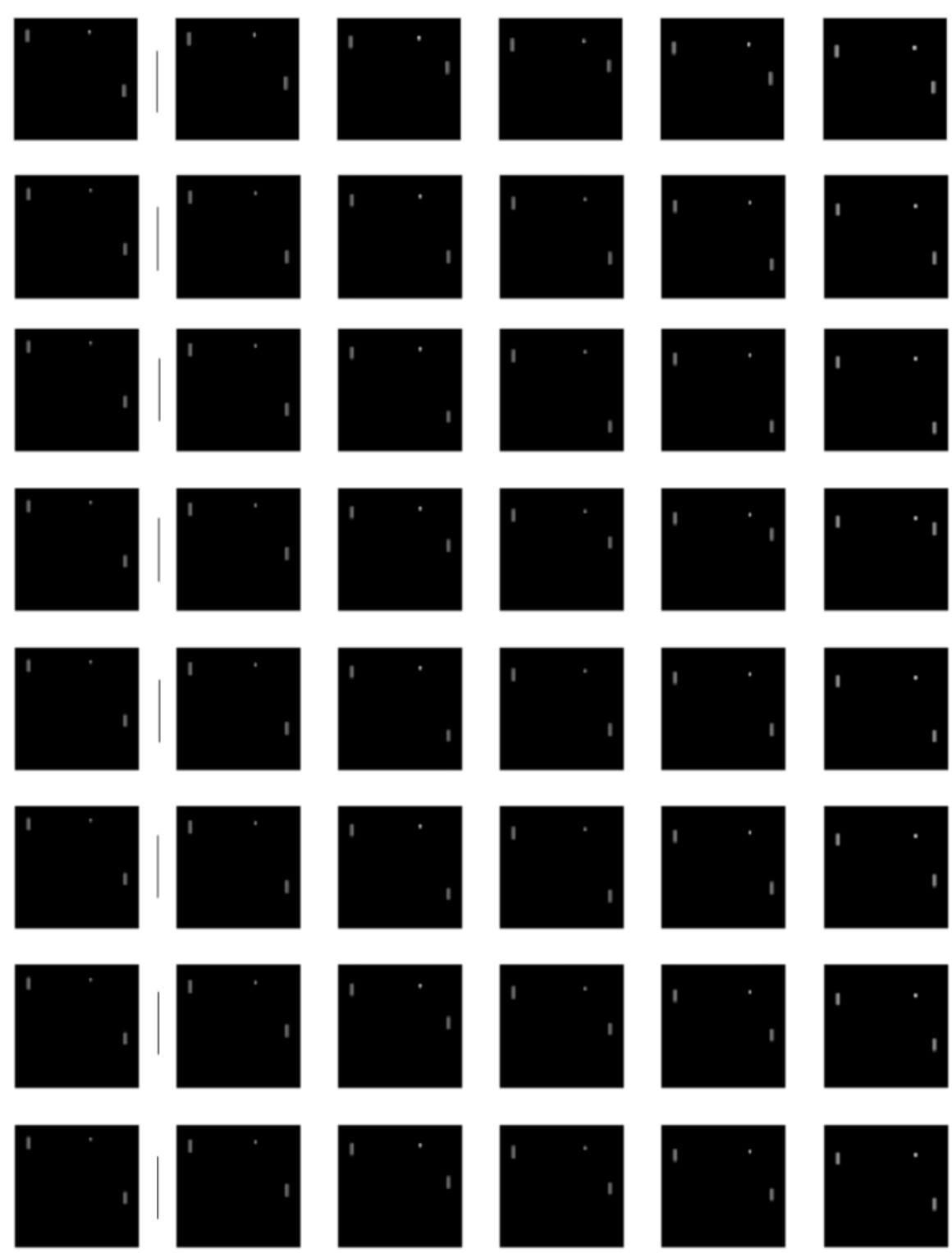

Figure 12: Eight 5-step roll-outs of the GDM on the Pong domain. Generated by sampling an initial state with 8 different 5-action length sequences.

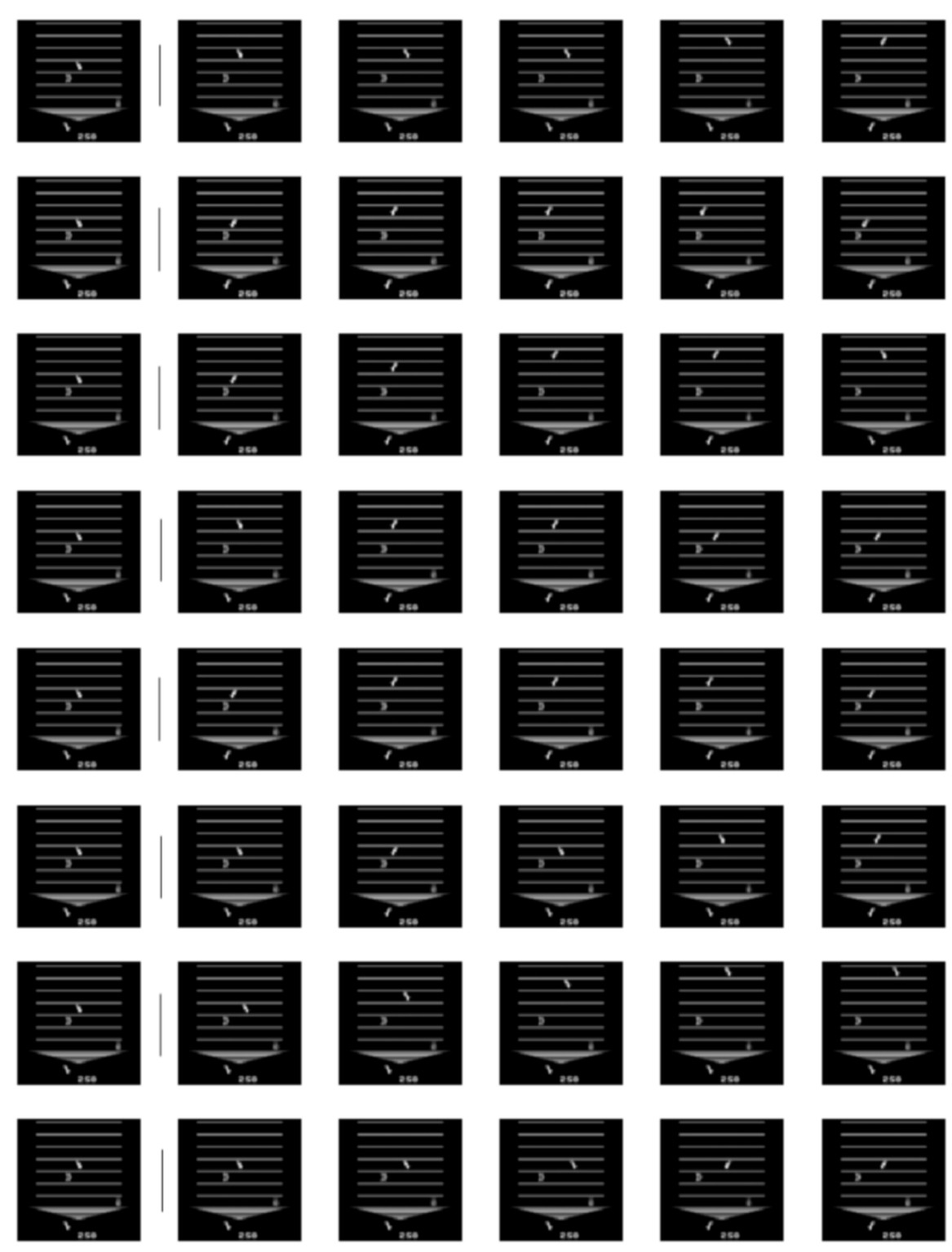

Figure 13: Eight 5-step roll-outs of the GDM on the Asterix domain. Generated by sampling an initial state with 8 different 5-action length sequences.

