# OpenReview forum: "Surprising Negative Results for Generative  Adversarial Tree Search "
_ICLR.cc/2019/Conference_

### Official Review · AnonReviewer3 · 2018-11-03
**The idea of the paper is interesting and it is valuable to share negative results but it would be beneficial if the paper would focus more on hypothesis evaluation in a more constraint environment.**

**Rating:** 6
**Confidence:** 2

**Review:**

This paper proposes to learn a dynamics model (state to pixels using Generative Adversarial Networks), use this model in conjunction with Monte Carlo Tree Search, model-free reinforcement learning (Q-learning) and a reward prediction network essentially combining model-free with model-based learning. The proposed approach is empirically evaluated on a small subset of the Atari games and theoretical analysis for the bias-variance tradeoff are presented.

It is highly appreciated that this paper presents an idea and discusses why the proposed approach does not result in high performance. This is very valuable and useful for the community. On a high level it would be very useful if Figure 1 would be show less examples but present them much larger since it is almost impossible to see anything in a printout. Further, the caption does not lend itself to understand the figure. Similarly Figure 2 would benefit from a better caption.

The first part of the discussion (7), the individual building blocks, should be mentioned much earlier in the paper. It would be further useful to add more related work on that abstraction level. This would help to communicate the main contribution of this paper very precisely.

On the discussion of negative results: It is very interesting that Dyna-Q does not improve the performance and the hypothesis for why this is the case seems reasonable. Yet, it would be very useful to actually perform an experiment in a better controlled environment for which e.g. the dynamics model is based on the oracle and assess the empirical effect of different MCTS horizons and rollout estimates. Further, this scenario would allow to further quantify the importance and the required “quality” of the different learning blocks.

In its current form the paper has theoretical contributions and experimental results which cannot be presented in the main paper due to space constraints. Albeit the appendix is already very extensive it would be very useful to structure it into the theoretical derivation and then one section per experiment with even more detail on the different aspects of the experiment. The story of the main paper would benefit from referencing the negative results more briefly and better analyzing the different hypothesis on toy like examples. Further, the introduction could be condensed in order to allow for more in detail explanations and discussions without repetition later on.

As argued in the paper it is clear that image generation is a very expensive simulation mechanism which for games like pong which depend on accurate modeling of small aspects of the image are in itself difficult. Therefore, again although really appreciated, the negative results should be summarized in the main paper and the hypothesis concluded better analyzed. The extensive discussion of hyper parameters and approaches for individual components could be in the appendix and the main paper focuses on the hypothesis analysis.

---

> ### Author Response · Authors · 2018-11-23
> **Response to AnonReviewer3**
>
> Thanks for the positive assessment and clear thoughtful feedback. We have improved the draft substantially per your feedback. Please find specific points below:
>
> *** Synthetic Examples***
> Per your suggestions, we implemented a controlled environment “Goldfish and gold bucket” environment highlighted in the discussion section, and evaluated the GATS algorithm with and without Dyna-Q in this synthetic environment. In this experiment, we give the agent access to the true environment, as suggested by the reviewer. Empirically, we find that GATS with short roll-outs (of lengths 1 and 2) consistently results in slower learning than vanilla DQN. Please see the graph of full results in Figure 2.
>
> *** The individual building blocks of GATS ***
> Per your feedback, we reorganized the paper to include the individual building blocks of GATS in the introduction. We also expand upon the building blocks in the related works section to better illustrate the ways the GATS framework can be extended. We believe that these improvements will help to highlight the general contributions of the insights of this paper for studying model-based and model-free reinforcement learning.
>
> *** Figures ***
> In the updated draft, we made Figure 1 and Figures 12-13 (in the appendix) significantly larger. The current version should be much easier to view, especially in print.
>
> *** Structure ***
> We have restructured the appendix to include the derivation of the theoretic result first, followed by individual empirical results. As suggested by the reviewer, we have added significant additional analysis of the controlled experiments mentioned above in the discussion and expanded upon the analysis of the negative results.

---

### Official Review · AnonReviewer2 · 2018-11-04
**Novel idea, but requires more work.**

**Rating:** 5
**Confidence:** 4

**Review:**

This paper presents Generative Adversarial Tree Search (GATS) that simulates trajectories for the Monte-Carlo Tree Search up to a certain depth. The GATS functions as a model that can simulate the dynamics of the system and predict rewards, effectively removing the need for a real simulator in the MCTS search.

They prove some favourable theoretical properties regarding the bias variance trade-off.  Specifically, they propose the model based model free trade-off which illustrates the trade-off between selecting longer rollouts, which increases the upper performance bound, and getting a more accurate Q estimate, which decreases the upper performance bound.

They also propose a pseudo count exploration bonus based on the inverse Wasserstein metric as the exploration strategy for GATS.

They observe that when tree-search rollouts are short, GATS fails to outperform DQN on 4 different games.

Quality:
It is unclear to me how you arrive at the result in Equation (9) of Appendix D. You have assumed in the second equation that the optimal action max_a Q(s,a) and max_a \hat{Q}(s,a) are the same action a. How do you arrive at this conclusion? Since \hat{Q} is an approximation of Q, why would the action a be the same?

Clarity:
The paper is fairly well written. There are many grammatical mistakes, but the overall message is more or less conveyed.

Originality:
It is original in the sense that a generative adversarial network is used as the model for doing the tree search. It is disappointing that this model does not yield better performance than the baseline and the theoretical results are questionable. I would like the authors to specifically address the theory in the rebuttal.

Significance:

While I appreciate negative results and there should be more papers like this,  I do think that this paper falls short in a couple of areas that I think the authors need to address. (1) As mentioned in quality, it is unclear to me that the theoretical derivation is correct. (2) The exploration bonus based on the inverse Wasserstein metric would add much value to the paper if it had an accompanying regret analysis (similar to UCB, for example, but adapted to the sequential MDP setting).

It appears in your transfer experiments that you do indeed train the GDM faster to adapt to the model dynamics, but it doesn’t appear to help your GATS algorithm actually converge to a good level of performance. Perhaps this paper should be re-written as a paper that focuses specifically on learning models that can easily transfer between domains with low sample complexity?

For the exploration bonus: If the authors added a regret analysis of the exploration count and can derive a bound of the number of times a sub-optimal action is chosen, then this could definitely strengthen the paper. This analysis could provide theoretical grounding and understanding for why their new exploration account makes sense, rather than basing it on empirical findings.

---

> ### Author Response · Authors · 2018-11-23
> **Response to AnonReviewer2**
>
> Thank you for this constructive review. We have improved the draft substantially to address your concrete concerns and are optimistic that the updated draft addresses your concerns sufficiently to warrant revisiting your assessment. We comment below briefly on the theoretical derivation, originality, and exploration.
>
> ***Theoretical Derivation***
> Regarding the reviewer’s concerns on the theoretical derivation in Eq. 9: We have added a detailed section to the derivation showing that the bound still holds when the optimal action max_a Q(s,a) and max_a \hat{Q}(s,a) are not the same action. Please refer to the updated draft directly and lemma 1 on Page 14. We have also revised the rest of the derivation to make this more clear.
>
> *** Negative Results ***
> We were also surprised that GATS does not yield better results on the studied task, but we believe both 1) that sufficiently surprising negative results hold scientific value and 2) that in particular, these results shed light on fundamental issues that arise when deploying MCTS together with a learned Q-function. To better highlight this surprising learning process, we included new experiments on a toy environment called “Goldfish and gold bucket”. Our empirical results in this controlled environment (see updated Fig. 2) demonstrate that even with perfect modeling, GATS with short rollouts can hurt performance, as seen in our Atari experiments.
>
>
> *** Inverse Wasserstein exploration***
> We agree with the reviewer that a proper regret analysis of our proposed inverse Wasserstein exploration method would be very insightful to the community. However, we point out that this analysis is not straightforward and might constitute a lengthy paper unto itself. While we are excited about this line of research, we left the regret bound analysis for future work.
>
> *** Domain transfer***
> As the reviewer notes, the goal of our domain transfer experiments was to show that the GDM model is powerful and, importantly, quick to adapt to changing model dynamics than the learned Q-function. The quality of the GDM is precisely what makes the negative results so surprising, since even powerful and accurate dynamics models may not result in improved results when MCTS and Q-learning are deployed together with short roll-outs, as in GATS. Thus, our results shed light on approaches that use model-based learning to improve domain transfer, which we hope to expand on in the future.

---

### Official Review · AnonReviewer1 · 2018-11-11
**Interesting paper with some missing depths in the analysis of the negative results**

**Rating:** 5
**Confidence:** 3

**Review:**

The submitted paper proposes GATS, a RL model combining model-free and model-bases reinforcement learning. Estimated models of rewards and dynamics are used to perform rollouts in MTCS without actually interacting with the true environment. The authors theoretically and empirically evaluate their approach for low depths rollouts. Empirically they show improved sample complexity on the Atari game Pong.

I think publishing negative research results is very important and should be done more often if we can learn from those results. But that is an aspect I feel this paper falls short at. I understand that the authors performed a great deal of effort trying to tune their model and evaluated many possible design choices, but they do not a provide a thorough investigation of the causes which make GATS "fail". I suggest that the authors try to understand the problems of MCTS with inaccurate models better with synthetic examples first. This could give insights into what the main sources of the problem are and how they might be circumvented. This would make the negative results much more insightful to the reader as each source of error is fully understood (e.g., beyond an error reate for predicting rewards which does not tell us about the distribution of errors which for example could have a big effect on the author's observations).

Another issue that needs further investigation is the author's "hypothesis on negative results". It would be great to experimentally underline the author's arguments. It is not trivial (at least to me) to fully see the "expected" interaction of learning dynamics and depths of rollouts. While MCTS should be optimal with any depths of rollouts given the true Q-function, the learning process seems more difficult to understand.

I would also like the authors to clarify one aspect of their theoretical analysis. e_Q is defined as the error in the Q-estimate for any single state and action in the main text. This appears to be inconsistent with the proof in the appendix, making the bound miss a factor which is exponential in H (all possible x_H that can be reached within H steps). This would change the properties of the bound quite significantly. Maybe I missed something, so please clarify.

Originality mainly comes from the use of GANs in MCTS and the theoretical analysis.

Strengths:
- Interesting research at the intersection of model-free and model-based research
- Lots of effort went into properly evaluating a wide range of possible design choices
- Mainly well written

Weaknesses:
- Missing depths in providing a deep understanding of why the author's expectations and empirical findings are inconsistent
- The authors use many tweaks and ideas to tune each component of their model making it difficult to draw conclusions about the exact contribution of each of these
- Error in the theoretical analysis (?)

Minor comment:
* The paper would benefit from improved and more self-contained figure captions.

---

> ### Author Response · Authors · 2018-11-23
> **Response to AnonReviewer1**
>
> Thank you for the detailed review and thoughtful suggestions. First, inspired by your suggestion to develop a synthetic example to demonstrate our negative results, we devised the “Goldfish and gold bucket” environment (described below). Additionally, in the latest version, we have addressed your concerns about theory (see, e.g., Page 14, Lemma 1). We describe the improvements to the draft below in detail.
>
> ***Synthetic Example***
>  We devised a simple grid-world based environment called “Goldfish and gold bucket”, where an agent must navigate an environment containing sharks (reward -1) and a gold bucket (reward 1).  We evaluated generative adversarial tree search (GATS) in the environment both with and without Dyna-Q. Interestingly, we observed that MCTS with short roll-out lengths consistently results in slower learning than vanilla DQN. Find the full quantitative results in Figure 2. We also released the implementation code of our synthetic study publicly.
>
> ***Theoretical issues***
> Thanks for your attention to detail. Regarding the e_Q term, in the original proof, we accidentally omitted the P(x_H | x, pi_r) term in the equation above Eq. 9. With this term, the sum is bounded by the max difference in the Q estimate for any given state, which is the e_Q term as pointed out by the reviewer. Additionally, per feedback from Reviewer 2, we clarified some steps in the derivation of the theorem, including a few (previously skipped) steps. Please find our detailed derivation in the updated paper following Lemma 1.
>
> Regarding the study of GATS performance with inaccurate models: We would like to bring the reviewer’s attention to the fact that the surprising result in this paper is that the GATS algorithm, even with an accurate model, results in a deterioration in the performance. The experimental study on the synthetic environment illustrates this phenomenon, even with a perfect model of the environment. In this synthetic study, GATS with limited depth (e.g. 1 and 2), underperformed a model learned with vanilla DQN.
>
> Regarding the optimality of MCTS: As the reviewer mentioned, MCTS using the true Q function is indeed optimal, but deploying MCTS with Q learning can have complex interactions. We believe that the new synthetic experiments on “Goldfish and gold bucket” help to better illuminate this complex learning process, which we first observed in our Atari experiments. In Proposition 1, we show that given a fixed estimated Q function, MCTS results in a better worst-case error in the Q estimation. But we would like to emphasize that these theoretical results do not guarantee better results. Our experiments indicate that deploying MCTS with Q learning can result in the learning of worse Q functions, which later is used in the leaf nodes. Moreover, Proposition 1 shows that the contribution in the error of Q estimation goes down exponentially in \gamma^H, but when \gamma is equal to 0.99 (a common choice for DRL in Atari games), we might not see much improvement with short rollouts.
>
> Regarding the figure captions: We appreciate the feedback. Already we have added a more detailed explanation in the captions in the updated draft and will continue to work to improve our exposition.

---

### Author Response · Authors · 2018-11-23
**General reply to reviewers and area chair**

First, we thank the reviewers for three detailed and thoughtful responses to our paper. We were glad to see that the reviewers found the approach interesting and that they appreciated our decision to submit a negative result for publication. While the original scores place the paper on the borderline, the reviewers made exceptionally specific requests for clarifications to the theory and additional extensive experiments to illustrate our negative results in a more controlled environment. Over the past few weeks, we have worked hard to improve the paper, clarifying out theoretical analysis. Per the reviewers’ suggestions, we also devised a toy environment, “Goldfish and gold bucket”, demonstrating our negative results in a controlled setting. We are grateful to the reviewers for their suggestions and hope that they will consider these substantial improvements when updating their reviews and scores. Please find specific replies to each reviewer in the respective threads.

---

### Meta-Review · Area_Chair1 · 2018-12-17
**A valuable direction, needs more systematic analysis into possible causes of negative results**

**Confidence:** 4
**Recommendation:** Reject

**Metareview:**

The paper addresses questions on the relationship between model-free and model-based reinforcement learning, in particular focusing on planning using learned generative models. The proposed approach, GATS, uses learned generative models for rollouts in MCTS, and provide theoretical insights that show a favorable bias-variance tradeoff. Despite this theoretical advantage, and high-quality models, the proposed approach fails to perform well empirically. This surprising negative results motivates the paper and providing insights on it is the main contribution.

Based on the initial submitted version, the reviewers positively emphasized the need to understand and publish important negative results. All reviewers and the AC appreciate the import role that such a contribution can bring to the research community. Reviewers also note the careful discussion of modeling choices for the generative models.

The reviewers also noted several potential weaknesses. Central were the need to better motivate and investigate the hypothesis proposed to explain the negative results. Several avenues towards a better understanding were proposed, and many of these were picked up by the authors in the revision and rebuttal. A novel toy domain "goldfish and gold bucket" was introduced for empirical analysis, and experiments there show that GATS can outperform DQN when a longer planning horizon is used.

The introduced toy domain provides additional insights into the relationship between planning horizon and GATS / MCTS performance. However, it does not address key questions around why the negative result is maintained. The authors hypothesize that the Q-value is less accurate in the GATS setting - this is something that can be empirically evaluated, but specific evidence for this hypothesis is not clearly shown. Other forms of analysis that could shed further light on why the specific negative result occurs could be to inspect model errors. For example, if generated frames are sorted by the magnitude of prediction errors - what are the largest mistakes? Could these cause learning performance to deteriorate?

The reviewers also raised several issues around the theoretical analysis, clarity (especially of captions) and structure - these were largely addressed by the revision. The concern that most strongly affected the final evaluation is the limited insight (and evidence) of the factors that influence performance of the proposed approach. Due to this, the consensus is to not accept the paper for publication at ICLR at this stage.